# A Domain Agnostic Measure for Monitoring and Evaluating GANs

**Paulina Grnarova**[*]
ETH Zurich

**Kfir Y. Levy**
Technion-Israel Institute of Technology

**Aurelien Lucchi**
ETH Zurich

**Nathanaël Perraudin**
Swiss Data Science Center

**Ian Goodfellow**

**Thomas Hofmann**
ETH Zurich

**Andreas Krause**
ETH Zurich

## Abstract

Generative Adversarial Networks (GANs) have shown remarkable results in modeling complex distributions, but their evaluation remains an unsettled issue. Evaluations are essential for: (i) relative assessment of different models and (ii) monitoring the progress of a single model throughout training. The latter cannot be determined by simply inspecting the generator and discriminator loss curves as they behave non-intuitively. We leverage the notion of duality gap from game theory to propose a measure that addresses both (i) and (ii) at a low computational cost. Extensive experiments show the effectiveness of this measure to rank different GAN models and capture the typical GAN failure scenarios, including mode collapse and non-convergent behaviours. This evaluation metric also provides meaningful monitoring on the progression of the loss during training. It highly correlates with FID on natural image datasets, and with domain specific scores for text, sound and cosmology data where FID is not directly suitable. In particular, our proposed metric requires no labels or a pretrained classifier, making it domain agnostic.

## 1 Introduction

In recent years, a large body of research has focused on practical and theoretical aspects of Generative adversarial networks (GANs) [9]. This has led to the development of several GAN variants [24, 2] as well as some evaluation metrics such as FID or the Inception score that are both data-dependent and dedicated to images. A *domain independent* quantitative metric is however still a key missing ingredient that hinders further developments.

One of the main reasons behind the lack of such a metric originates from the nature of GANs that implement an adversarial game between two players, namely a generator and a discriminator. Let us denote the data distribution by $p_{\text{data}}(\mathbf{x})$, the model distribution by $p_{\mathbf{u}}(\mathbf{x})$ and the prior over latent variables by $p_{\mathbf{z}}$. A probabilistic discriminator is denoted by $D_{\mathbf{v}} : \mathbf{x} \mapsto [0; 1]$ and a generator by $G_{\mathbf{u}} : \mathbf{z} \mapsto \mathbf{x}$. The GAN objective is:

$$\min_{\mathbf{u}} \max_{\mathbf{v}} M(\mathbf{u}, \mathbf{v}) = \frac{1}{2}\mathbb{E}_{\mathbf{x} \sim p_{\text{data}}} \log D_{\mathbf{v}}(\mathbf{x}) + \frac{1}{2}\mathbb{E}_{\mathbf{z} \sim p_{\mathbf{z}}} \log(1 - D_{\mathbf{v}}(G_{\mathbf{u}}(\mathbf{z}))). \quad (1)$$

Each of the two players tries to optimize their own objective, which is exactly balanced by the loss of the other player, thus yielding a two-player zero-sum minimax game. The minimax nature of the objective and the use of neural networks as players make the process of learning a generative model challenging. We focus our attention on two of the central open issues behind these difficulties and how they translate to a need for an assessment metric.

---

[*]Correspondence to `paulina.grnarova@inf.ethz.ch`

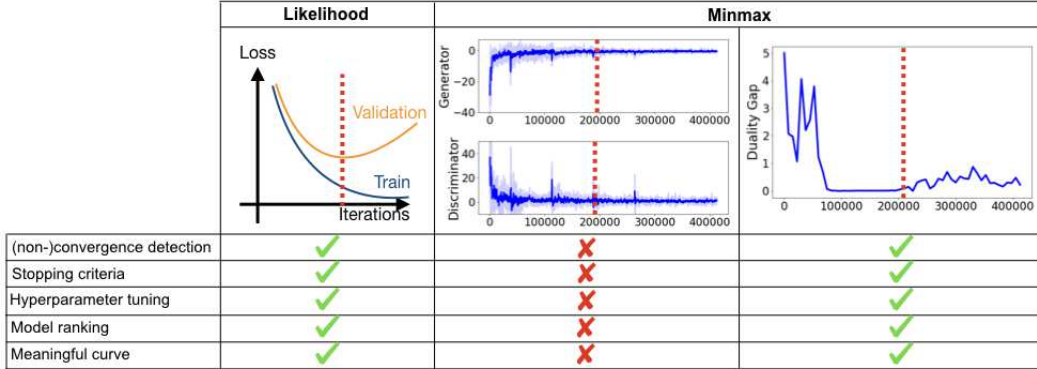

Figure 1: Comparison of information obtained by different metrics for likelihood and minmax-based models. The red dashed line corresponds to the optimal point for stopping the training.

**i) Convergence metric**    The need for an adequate convergence metric is especially relevant given the difficulty of training GANs: current approaches often fail to converge [31] or oscillate between different modes of the data distribution [21]. The ability of reliably detecting non-convergent behavior has been pointed out as an open problem in many previous works, e.g., by [20] as a stepping stone towards a deeper analysis as to which GAN variants converge. Such a metric is not only important for driving the research efforts forward, but from a practical perspective as well. Deciding when to stop training is difficult as the curves of the discriminator and generator losses oscillate (see Fig. 1) and are non-informative as to whether the model is improving or not [2]. This is especially troublesome when a GAN is trained on non-image data in which case one might *not* be able to use visual inspection or FID/Inception scores as a proxy.

**ii) Evaluation metric**    Another key problem we address is the relative comparison of the learned generative models. While several evaluation metrics exist, there is no clear consensus regarding which metric is the most appropriate. Many metrics achieve reasonable discriminability (i.e., ability to distinguish generated samples from real ones), but also tend to have a high computational cost. Some popular metrics are also specific to image data. We refer the reader to [3] for an in-depth discussion of the merits and drawbacks of existing evaluation metrics.

In more traditional likelihood-based models, the train/test curves do address the problems raised in **i)** and **ii)**. For GANs, the generator/discriminator curves (see Fig. 1) are however largely uninformative due to the minimax nature of GANs where both players can undo each other's progress.

In this paper, we leverage ideas from game theory to propose a simple and computationally efficient metric for GANs. Our approach is to view GANs as a zero-sum game between a generator $G$ and discriminator $D$. From this perspective, "solving" the game is equivalent to finding an *equilibrium*, i.e., a pair $(G^*, D^*)$ such that no side may increase its utility by unilateral deviation. A natural metric for measuring the sub-optimality (w.r.t. an equilibrium) of a given solution $(G, D)$ is the *duality gap* [33, 22]. We therefore suggest to use it as a metric for GANs akin to a test loss in the likelihood case (See Fig. 1 - duality gap[2]).

There are several important issues that we address in order to make the *duality gap* an appropriate and practical metric for GANs. Our contributions include the following:

- We show that the *duality gap* allows to assess the similarity between the generated data and true data distribution (see Theorem 1).
- We show how to appropriately estimate the *duality gap* in the typical machine learning scenario where our access to the GAN learning objective is only through samples.
- We provide a computationally efficient way to estimate the *duality gap* during training.
- In scenarios where one is interested in assessing the quality of the learned generator, we show how to use a related metric – the minimax loss – that takes only the generator into consideration in order to detect mode collapse and measure sample quality.

- We extensively demonstrate the effectiveness of these metrics on a range of datasets, GAN variants and failure modes. Unlike the FID or Inception score that require labelled data or a domain dependent classifier, our metrics are *domain independent* and do not require labels.

**Related work.** While several evaluation metrics have been proposed [31, 30, 12, 18], previous research has pointed out various limitations of these metrics, thus leaving the evaluation of GANs as an unsettled issue [20]. Since the data log-likelihood is commonly used to train generative models, it may appear to be a sensible metric for GANs. However, its computation is often intractable and [32] also demonstrate that it has severe limitations as it might yield low visual quality samples despite of a high likelihood. Perhaps the most popular evaluation metric for GANs is the inception score [31] that measures both diversity of the generated samples and discriminability. While diversity is measured as the entropy of the output distribution, the discriminability aspect requires a pretrained neural network to assign high scores to images close to training images. Various modifications of the inception score have been suggested. The Frechet Inception Distance (FID) [12] models features from a hidden layer as two multivariate Gaussians for the generated and true data. However, the Gaussian assumption might not hold in practice and labelled data is required in order to train a classifier. Without labels, transfer learning is possible to datasets under limited conditions (i.e., the source and target distributions should not be too dissimilar). In [26], two metrics are introduced to evaluate a single model playing against past and future versions of itself, as well as to measure the aptitude of two different fully trained models. In some way, this can be seen as an approximation of the minimax value we advocate in this paper, where instead of doing a full-on optimization in order to find the best adversary for the fixed generator, the search space is limited to discriminators that are snapshots from training, or discriminators trained with different seeds.

The ideas of duality and equilibria developed in the seminal work of [33, 22] have become a cornerstone in many fields of science, but are relatively unexplored for GANs. Some exceptions are [5, 10, 8, 13] but these works do not address the problem of evaluation. Closer to us, game theoretic metrics were previously mentioned in [25], but without a discussion addressing the stochastic nature and other practical difficulties of GANs, thus not yielding a practical applicable method. We conclude our discussion by pointing out the vast literature on duality used in the optimization community as a convergence criterion for min-max saddle point problems, see e.g. [23, 14]. Some recent work uses Lagrangian duality in order to derive an objective to train GANs [4] or to dualize the discriminator, therefore reformulating the saddle point objective as a maximization problem [17]. A similar approach proposed by [7] uses the dual formulation of Wasserstein GANs to train the decoder. Although we also make use of duality, there are significant differences. Unlike prior work, our contribution does not relate to *optimising GANs*. Instead, we focus on establishing that the duality gap acts as a proxy to measure convergence, which we do theoretically (Th. 1) as well as empirically, the latter requiring a new efficient estimation procedure discussed in Sec. 3.

## 2   Duality Gap as Performance Measure

Standard learning tasks are often described as (stochastic) optimization problems; this applies to common Deep Learning scenarios as well as to classical tasks such as logistic and linear regression. This formulation gives rise to a natural performance measure, namely the test loss[3]. In contrast, GANs are formulated as (stochastic) zero-sum games. Unfortunately, this fundamentally different formulation does not allow us to use the same performance metric. In this section, we describe a performance measure for GANs, which naturally arises from a game theoretic perspective. We start with a brief overview of zero-sum games, including a description of the *Duality gap* metric.

A zero-sum game is defined by two players $\mathcal{P}_1$ and $\mathcal{P}_2$ who choose a decision from their respective decision sets $\mathcal{K}_1$ and $\mathcal{K}_2$. A game objective $M : \mathcal{K}_1 \times \mathcal{K}_2 \mapsto \mathbb{R}$ sets the utilities of the players. Concretely, upon choosing a pure strategy $(\mathbf{u}, \mathbf{v}) \in \mathcal{K}_1 \times \mathcal{K}_2$ the utility of $\mathcal{P}_1$ is $-M(\mathbf{u}, \mathbf{v})$, while the utility of $\mathcal{P}_2$ is $M(\mathbf{u}, \mathbf{v})$. The goal of either $\mathcal{P}_1$/$\mathcal{P}_2$ is to maximize their worst case utilities:

$$\min_{\mathbf{u} \in \mathcal{K}_1} \max_{\mathbf{v} \in \mathcal{K}_2} M(\mathbf{u}, \mathbf{v}) \quad \textbf{(Goal of } \mathcal{P}_1 \textbf{)}, \quad \max_{\mathbf{v} \in \mathcal{K}_2} \min_{\mathbf{u} \in \mathcal{K}_1} M(\mathbf{u}, \mathbf{v}) \quad \textbf{(Goal of } \mathcal{P}_2 \textbf{)} \tag{2}$$

This formulation raises the question of whether there exists a solution $(\mathbf{u}^*, \mathbf{v}^*)$ to which both players may jointly converge. The latter only occurs if there exists $(\mathbf{u}^*, \mathbf{v}^*)$ such that neither $\mathcal{P}_1$ nor $\mathcal{P}_2$ may

increase their utility by unilateral deviation. Such a solution is called a *pure equilibrium*, formally,

$$\max_{\mathbf{v} \in \mathcal{K}_2} M(\mathbf{u}^*, \mathbf{v}) = \min_{\mathbf{u} \in \mathcal{K}_1} M(\mathbf{u}, \mathbf{v}^*) \quad \textbf{(Pure Equilibrium).}$$

While a pure equilibrium does not always exist, the seminal work of [22] shows that an extended notion of equilibrium always does. Specifically, there always exists a distribution $\mathcal{D}_1$ over elements of $\mathcal{K}_1$, and a distribution $\mathcal{D}_2$ over elements of $\mathcal{K}_2$, such that the following holds,

$$\max_{\mathbf{v} \in \mathcal{K}_2} \mathbb{E}_{\mathbf{u} \sim \mathcal{D}_1} M(\mathbf{u}, \mathbf{v}) = \min_{\mathbf{u} \in \mathcal{K}_1} \mathbb{E}_{\mathbf{v} \sim \mathcal{D}_2} M(\mathbf{u}, \mathbf{v}) \quad \textbf{(MNE).}$$

Such a solution is called a *Mixed Nash Equilibrium (MNE)*. This notion of equilibrium gives rise to the following natural performance measure of a given pure/mixed strategy.

**Definition 1** (Duality Gap). *Let $\mathcal{D}_1$ and $\mathcal{D}_2$ be fixed distributions over elements from $\mathcal{K}_1$ and $\mathcal{K}_2$ respectively. Then the duality gap DG of $(\mathcal{D}_1, \mathcal{D}_2)$ is defined as follows,*

$$\mathrm{DG}(\mathcal{D}_1, \mathcal{D}_2) := \max_{\mathbf{v} \in \mathcal{K}_2} \mathbb{E}_{\mathbf{u} \sim \mathcal{D}_1} \mathrm{M}(\mathbf{u}, \mathbf{v}) - \min_{\mathbf{u} \in \mathcal{K}_1} \mathbb{E}_{\mathbf{v} \sim \mathcal{D}_2} \mathrm{M}(\mathbf{u}, \mathbf{v}). \tag{3}$$

*Particularly, for a given* pure strategy $(\mathbf{u}, \mathbf{v}) \in \mathcal{K}_1 \times \mathcal{K}_2$ *we define,*

$$\mathrm{DG}(\mathbf{u}, \mathbf{v}) := \max_{\mathbf{v}' \in \mathcal{K}_2} \mathrm{M}(\mathbf{u}, \mathbf{v}') - \min_{\mathbf{u}' \in \mathcal{K}_1} \mathrm{M}(\mathbf{u}', \mathbf{v}). \tag{4}$$

Two well-known properties of the duality gap are that it is always non-negative and is exactly zero in (mixed) Nash Equilibria. These properties are very appealing from a practical point of view, since it means that the duality gap gives us an immediate handle for measuring convergence.

Next we illustrate the usefulness of the duality gap metric by analyzing the ideal case where both $G$ and $D$ have unbounded capacity. The latter notion introduced by [9] means that the generator can represent any distribution, and the discriminator can represent any decision rule. The next proposition shows that in this case, as long as $G$ is not equal to the true distribution then the duality gap is always positive. In particular, we show that the duality gap is at least as large as the Jensen-Shannon divergence between true and fake distributions. We also show that if $G$ outputs the true distribution, then there exists a discriminator such that the duality gap (DG) is zero. See a proof in the Appendix.

**Theorem 1** (DG and JSD). *Consider the GAN objective in Eq. [1], and assume that the generator and discriminator networks have unbounded capacity. Then the duality gap of a given fixed solution $(G_{\mathbf{u}}, D_{\mathbf{v}})$ is lower bounded by the Jensen-Shannon divergence between the true distribution $p_{\mathrm{data}}$ and the fake distribution $q_{\mathbf{u}}$ generated by $G_{\mathbf{u}}$, i.e. $\mathrm{DG}(\mathbf{u}, \mathbf{v}) \geq \mathrm{JSD}(p_{\mathrm{data}} \,\|\, q_{\mathbf{u}})$. Moreover, if $G_{\mathbf{u}}$ outputs the true distribution, then there exists a discriminator $D_{\mathbf{v}}$ such that $\mathrm{DG}(G_{\mathbf{u}}, D_{\mathbf{v}}) = 0$.*

Note that different GAN objectives are known to be related to other types of divergences [24], and we believe that the Theorem above can be generalized to other GAN objectives [2, 11].

## 3 Estimating the Duality Gap for GANs

**Appropriately estimating the duality gap from samples.** Supervised learning problems are often formulated as stochastic optimization programs, meaning that we may only access estimates of the expected loss by using samples. One typically splits the data into training and test sets [4]. The training set is used to find a solution whose quality is estimated using a separate test set (which provides an unbiased estimate of the true expected loss). Similarly, GANs are formulated as stochastic zero-sum games (Eq. [1]) but the issue of evaluating the duality gap metric is more delicate. This is because we have three phases in the evaluation: **(i)** training a model $(\mathbf{u}, \mathbf{v})$, **(ii)** finding the worst case discriminator/generator, $\mathbf{v}_{\mathrm{worst}} \leftarrow \arg \max_{\mathbf{v} \in \mathcal{K}_2} M(\mathbf{u}, \mathbf{v})$, and $\mathbf{u}_{\mathrm{worst}} \leftarrow \arg \min_{\mathbf{u} \in \mathcal{K}_1} M(\mathbf{u}, \mathbf{v})$, and **(iii)** computing the duality gap by estimating: $\mathrm{DG} := \mathrm{M}(\mathbf{u}, \mathbf{v}_{\mathrm{worst}}) - \mathrm{M}(\mathbf{u}_{\mathrm{worst}}, \mathbf{v})$. Now since we do not have direct access to the expected objective, one should use different samples for each of the three mentioned phases in order to maintain an unbiased estimate of the expected duality gap. Thus we split our dataset into three disjoint subsets: a training set, *an adversary finding set*, and a test set which are respectively used in phases **(i)**, **(ii)** and **(iii)**.

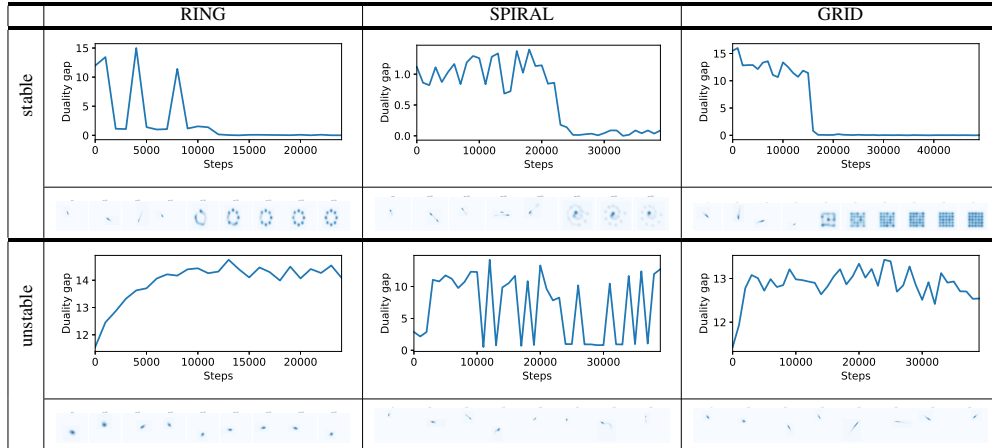

Figure 2: Progression of duality gap (DG) throughout training and heatmaps of generated samples.

**Minimax Loss as a metric for evaluating generators.** For all experiments, we report both the duality gap (DG) and the minimax loss $M(\mathbf{u}, \mathbf{v}_{\mathrm{worst}})$. The latter is the first term in the expression of the DG and intuitively measures the 'goodness' of a generator $G_{\mathbf{u}}$. If $G_{\mathbf{u}}$ is optimal and covers $p_{\mathrm{data}}$, the minimax loss achieves its optimal value as well. This happens when $D_{\mathbf{v}_{\mathrm{worst}}}$ outputs 0.5 for both the real and generated samples. Whenever the generated distribution does not cover the entire support of $p_{\mathrm{data}}$ or compromises the sample quality, this is detected by $D_{\mathbf{v}_{\mathrm{worst}}}$ and hence, the minimax loss increases. This makes it a compelling metric for detecting mode collapse and evaluating sample quality. Note that in order to compute this metric one only needs a batch of generated samples, i.e. the generator can be used as a black-box. Hence, this metric is not limited to generators trained as part of a GAN, but can instead be used for any generator that can be sampled from.

**Practical and efficient estimation of the duality gap for GANs.** In practice, the metrics are computed by optimizing a separate generator/discriminator using a gradient based algorithm. To speed up the optimization, we initialize the networks using the parameters of the adversary at the step being evaluated. Hence, if we are evaluating the GAN at step $t$, we train $\mathbf{v}_{\mathrm{worst}}$ for $\mathbf{u}_t$ and $\mathbf{u}_{\mathrm{worst}}$ for $\mathbf{v}_t$ by using $\mathbf{v}_t$ as a starting point for $\mathbf{v}_{\mathrm{worst}}$ and analogously, $\mathbf{u}_t$ as a starting point for $\mathbf{u}_{\mathrm{worst}}$ for a number of fixed steps. We also explored further approximations of DG, where instead of using optimization to find $\mathbf{v}_{\mathrm{worst}}$ and $\mathbf{u}_{\mathrm{worst}}$, we limit the search space to a set of discriminators and generators stored as snapshots throughout the training, similarly to [26] (see results in Appendix C.5). In Appendix D we include an in-depth analysis of the quality of the approximation of the DG and how it compares to the true theoretical DG.

**Non-negativity DG.** While DG is non-negative in theory, this might not hold in practice since we only find approximate $(\mathbf{u}_{\mathrm{worst}}, \mathbf{v}_{\mathrm{worst}})$. Nevertheless, the practical scheme that we describe above makes sure that we do not get negative values for DG in practice. We elaborate on this in AppendixB.

## 4 Experimental results

We carefully design a series of experiments to examine commonly encountered failure modes of GANs and analyze how this is reflected by the two metrics. Specifically, we show the sensitivity of the duality gap metric to (non-) convergence and the susceptibility of the minimax loss to reflect the sample quality. Further details and additional extensive experiments can be found in the Appendix. Note that our goal is *not* to provide a rigorous comparative analysis between different GAN variants, but to show that both metrics capture properties that are particularly useful to monitor training.

### 4.1 Mixture of Gaussians

We train a vanilla GAN on three toy datasets with increasing difficulty: a) RING: a mixture of 8 Gaussians, b) SPIRAL: a mixture of 20 Gaussians and c) GRID: a mixture of 25 Gaussians. As the true data distribution is known, this setting allows for tracking convergence.

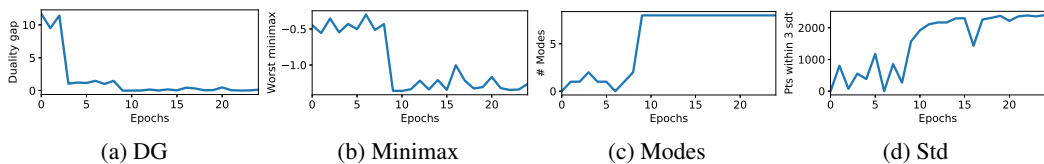

(a) DG        (b) Minimax        (c) Modes        (d) Std

Figure 3: DG, minimax, modes covered and std. Tab. 7 in App. shows Pearson correlation between the metrics.

**Duality gap and convergence.** Our first goal is to illustrate the relation between convergence and the duality gap. To that end, we analyze the progression of DG throughout training in stable and unstable settings. One common problem of GANs is *unstable mode collapse*, where the generator alternates between generating different modes. We simulate such instabilities and compare them against successful GANs in Fig. 2. The gap goes to zero for all stable models after convergence to the true data distribution. Conversely, unstable training is reflected both in terms of the large value reached by DG as well as its trend over iterations (e.g., large oscillations and an increasing trend indicate unstable behavior). Thus the duality gap is a powerful tool for *monitoring the training and detecting unstable collapse*.

**Minimax loss reflects sample quality.** As previously argued, another useful metric to look at is the minimax loss which focuses solely on the generator. For the toy datasets, we measure the sample quality using *(i)* the number of covered modes and *(ii)* the number of generated samples that fall within 3 standard deviations of the modes (std). Fig. 3 shows significant anti-correlation, which indicates *the minimax loss can be used as a proxy for determining the overall sample quality*.

### 4.2 Duality gap and stable mode collapse

The previous experiment shows how *unstable* mode collapse is captured by DG - the trend is unstable and is typically within a high range. We are now interested in the case of *stable mode collapse*, where the model does converge, but only to a subset of the modes.

We train a GAN on MNIST where the generator collapses to generating from only one class and does not change the mode as the number of training steps increases. Fig. 4 shows the DG curve. The trend is flat and stable, but the value of DG is not zero, thus showing that *looking at both the trend and value of the DG is helpful for detecting stable mode collapse as well*.

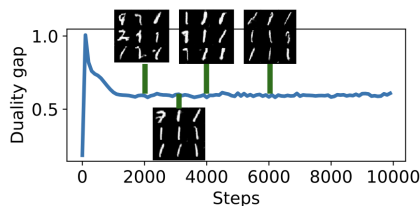

Figure 4: DG evolution detects stable mode collapse.

### 4.3 The trend of the duality gap progression curve

We now analyze the trend of the DG curves. The plots (see Fig. 2) show it does not always monotonically decrease throughout training as we do observe non-smooth spikes. This raises the question as to whether these spikes are the result of the instabilities of the training or due to the metric itself? To address this, we train a GAN on a 2D submanifold Gaussian mixture embedded in 3D space. Such a setting captures a commonly encountered GAN failure as this mixture is degenerate with respect to the base measure defined in ambient space due to the lack of fully dimensional support.

It has been shown [29] that an unregularized GAN collapses in every run after 50K iterations (see Fig. 5 - right) because of the focus of the discriminator on smaller differences between the true and generated samples, whereas the training of a regularized version can essentially avoid collapsing even well beyond 200K iterations (see Fig. 5 - left). Thus we would expect the DG curves for the two settings to show different trends, which is indeed what we observe. For the unregularized version, DG decreases to small values and is stable until 40K steps, after which it starts increasing, reflecting the collapse of the generator. A practitioner looking at the DG curve can thus learn that (i) the training should be stopped between 20-40K steps and (ii) there is a collapse in the training (information that is especially valuable when the generated data is of non-image type or cannot be visualised). Conversely, the DG trend for the regularized version is stable and very quickly converges to values

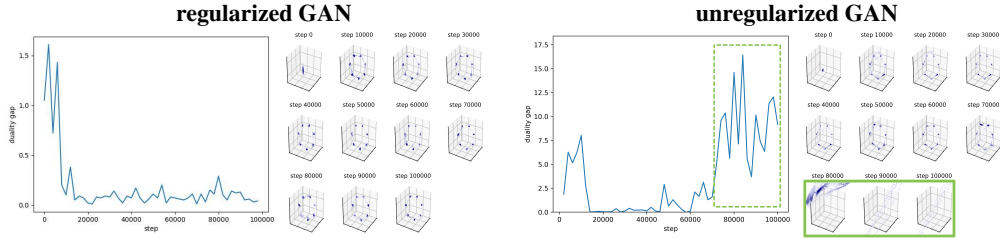

Figure 5: DG progression for a regularized (left) and unregularized GAN (right). Generated samples are shown for various steps. DG is able to capture instabilites (see green box).

close to zero, which reflects the improved quality. This suggests that *the usage of DG opens avenues to further understand and compare the effects of various regularizers*. Note that in [29] different levels of the regularizer were only visually compared due to the lack of a proper metric.

## 4.4 Comparison with image-specific criteria

We further analyze the sensitivity of the minimax loss to various changes in the sample quality for natural images that fall broadly in two categories: *(i)* mode sensitivity and *(ii)* visual sample quality. We compare against the commonly used Inception Score (INC) and Frechet Inception Distance (FID). Both metrics use the generator as a black-box through sampling. We follow the same setup for the evaluation of minimax and use the GAN zero-sum objective. Note that changing the objective to WGAN formulation makes it closely related to the Wasserstein critic [2].

**Sensitivity to modes.** As natural images are inherently multimodal, the generated distribution commonly ignores some of the true modes, which is a phenomenon known as *mode dropping*. Another common issue is *intra-mode collapse* that occurs when the generator is generating from all modes, but there is no variety within a mode. We then turn to *mode invention* where the generator creates non-existent modes. Fig. 18 in the Appendix shows the trends of all metrics for various degrees of mode dropping, invention and intra-mode collapse, where a class label is considered a mode. INC is unable to detect both intra-mode collapse and invented modes. On the other hand, *both FID and minimax loss exhibit desirable sensitivity to various mode changing*.

**Sample quality.** We study the metrics' ability to detect compromised sample quality by distorting real images using Gaussian noise, blur and swirl at an increasing intensity. As shown in Fig. 19 in the Appendix, *all metrics, including minimax, detect different degrees of visual sample quality*.

In Appendix C.3.1 we also show that the metric is *computationally efficient to be used in practice*. Tab. 1 gives a summary of all the results.

| Poperty\Metric | INC | FID | minimax |
|---|---|---|---|
| Sensitivity to mode collapse | moderate | high | high |
| Sensitivity to mode invention | low | high | high |
| Sensitivity to intra-mode collapse | low | high | high |
| Sensitivity to visual quality and transformations | moderate | high | high |
| Computational: Fast | yes | yes | yes |
| Computational: Needs labeled data or a pretrained classifier | yes | yes | no |
| Computational: Can be applied to any domain without change | no | no | yes |

Table 1: Comparison of INC, FID and minimax on various properties.

## 4.5 DG as a measure for the image domain

The previous section illustrates the metrics have desirable properties that makes them effective in capturing different failure modes in terms of image-specific criteria. Since generating images is one of the most common use cases of GANs, we further explore the usefulness of the DG on generating faces through a ProgGAN trained on CelebA. Figure 6 shows that unlike the GAN losses, the DG

trend can capture the progress, which is also in agreement with the trend of the largest singular values of the convolutional layers of G and D.

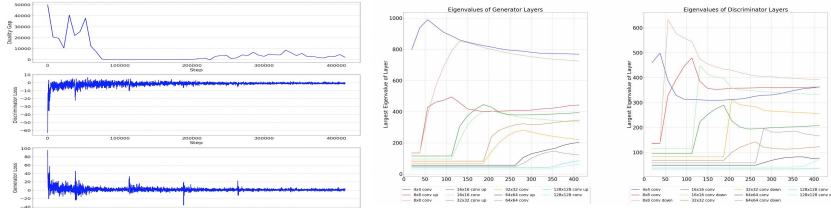

Figure 6: ProgGAN trained on CelebA: (left) losses vs. DG; (right) largest singular values of the conv layers

## 4.6 Generalization to other domains and GAN losses

These experiments test the ability of the two metrics to adapt to a different GAN loss formulation (WGAN-GP [11] and SeqGAN [34]), as well as other domains (cosmology, audio, text).

**N-body simulations in cosmology.** We consider the field of observational cosmology that relies on computationally expensive simulations with very different statistics from natural images. In an attempt to reduce this burden [28] trained a WGAN-GP to replace the traditional N-body simulators, relying on three statistics to assess the quality of the generated samples: mass histrogram, peak count and power spectral density. A random selection of real and generated samples shown in Fig. 21 in the Appendix demonstrate the high visual quality achieved by the generator.

We evaluate the agreement between the statistics of the real and generated samples using the squared norm of the statistics differences (lower scores are therefore better). In Fig. 7, we show the evolution of the scores corresponding to the three statistics as well as DG. We observe a strong correlation, especially between the peaks. Furthermore, it seems that the duality gap takes all the statistics into account. In Tab. 2, we observe a strong empirical correlation between the duality gap, the minimax value and the cosmological scores. We also observe that the FID is

|              | Mass hist. | Peak hist. | PSD  |
| ------------ | ---------- | ---------- | ---- |
| Dual gap     | 0.53       | 0.38       | 0.71 |
| Minmax value | 0.66       | 0.51       | 0.75 |
| FID          | 0.40       | 0.21       | 0.34 |

Table 2: Pearson correlation between cosmological scores (mass histogram, peak histogram and Power Spectral Density (PSD)) and metrics (dual gap, minimax and FID).

significantly less correlated that the duality gap, which is explained by the fact that the images we use are not natural images and hence that the statistics of the Inception Network are not suitable to evaluate the quality of cosmological samples.

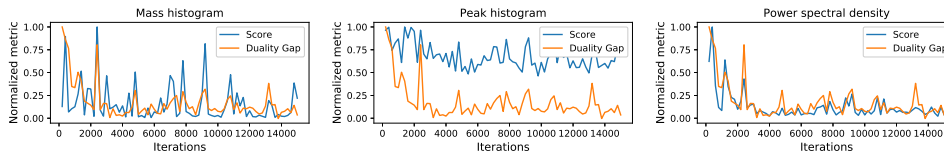

Figure 7: DG and cosmo-score evolution. DG strongly correlates with all 3 scores.

**Audio Time-Frequency consistency.** Generating an audio waveform is a challenging problem as it requires an agreement between scales from the range of milliseconds to tens of seconds. To overcome this challenge, one may use more powerful and intuitive features such as a Time-Frequency (TF) representation. Nevertheless, in order to obtain a natural listening experience, a TF representation needs to be consistent, i.e., there must exist an audio signal which leads to this TF representation. [19] define a measure estimating the consistency of a representation and use it to assess the convergence of their GAN. In Fig. 8, we show the evolution the measure and of DG and minimax. We observe a clear correlation, especially with the minimax value, which is expected as both the consistency measure and minimax evaluate only the generated samples.

**Text generation.** Another challenging modality for generative models is text. SeqGAN [34] is a GAN-based sequence framework evaluated by negative log-likelihood: *nll-oracle* and *nll-test*. The

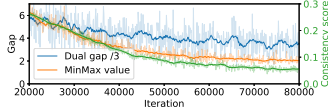

Figure 8: Evolution of the consistency measure vs. the DG.

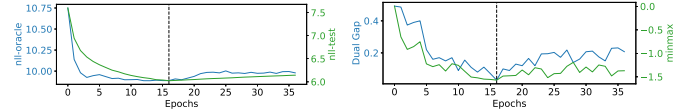

Figure 9: Evolution of nll-oracle and nll-test (left) vs. DG and minimax (right). The dashed line represents the optimal stopping point.

first metric computes the likelihood of generated data against an oracle, whereas the second makes use of the generator and the likelihood of real test data. Fig. 9 shows DG and minimax correlate well with both metrics. Moreover, both DG and minimax can determine the optimal stopping point.

## 4.7 Comparison of different models

So far, we have seen that our estimation of DG generalizes to various loss functions and data distributions. We now turn our attention to evaluating the generalization abilities to models computed from different classes (i.e. neural networks with different architectures). We propose to do this evaluation by taking the original GAN minmax objective as a reference. The reasoning

|           | FID/DG | FID/mm | INC/DG | INC/mm |
|-----------|--------|--------|--------|--------|
| NS        | 0.32   | 0.88   | -0.34  | -0.87  |
| NS + SN   | 0.93   | 0.85   | -0.93  | -0.91  |
| NS + SN + GP | 0.94 | 0.94   | -0.91  | -0.95  |
| WGAN + GP | 0.87   | 0.90   | -0.87  | 0.91   |

Table 3: Pearson correlation between FID and INC with DG and minimax (mm) for different GAN variants.

behind this choice is that a better discriminator would always be fooled less by a worst generator, irrespective of how it was trained. The analogue holds for the generator as well, resulting in a lower DG/minimax value on a selected objective.

To that end, we compare different commonly used ResNet based GAN variants on Cifar10: a GAN using the non saturating update rule (NS), spectrally normalized GAN (NS + SN), spectrally normalized GAN with the addition of a gradient penalty (NS + SN +

|                 | FID | | INC | | DG | | mm | |
|-----------------|-------|------|-------|------|-------|------|-------|------|
|                 | score | rank | score | rank | score | rank | score | rank |
| NS              | 28.65 | 3    | 7.1   | 3    | 2.36  | 3    | -1.07 | 3    |
| NS+SN           | 22.25 | 1    | 7.78  | 1    | 0.21  | 1    | -1.31 | 1    |
| NS+SN+GP        | 23.46 | 2    | 7.75  | 2    | 0.23  | 2    | -1.30 | 2    |
| WGAN+GP         | 72.23 | 4    | 4.93  | 4    | 4.53  | 4    | -0.2  | 4    |
| compute time *(s)* | 120.50 | | 47.33 | | 27.22 | | 7.38 | |

Table 4: Scores and ranking for various GAN models using different metrics. The final row gives the computation time in seconds.

GP) and a WGAN with gradient penalty (WGAN GP). We use the optimal hyperparameters suggested in [16]. Table 3 shows the Pearson correlation between DG and the minimax metric against FID and the Inception score (INC), which are known to work well on this dataset. We find that the minimax metric always highly correlates with both FID and INC. This is expected as (i) minimax evaluates the generator only, just as FID and INC that do not take the discriminator into consideration and (ii) as previously shown minimax is sensitive to mode changes and sample quality. Interestingly, DG also correlates highly whenever the discriminator is properly regularized. The level of correlation is however reduced for the unregularized variants. We hypothesise this is due to instabilities in the training, for which the generator might be improving, while the discriminator is becoming worse.

Tab. 4 shows the ranking of the models is the same for all four metrics, suggesting that *DG/minimax is a sensible choice for comparing different GAN models*.

## 5 Conclusion

We propose a domain agnostic evaluation measure for GANs that relies on the duality gap (DG) and upper bounds the JS divergence between real and generated data. This measure allows for meaningful monitoring of the progress made during training, which was lacking until now. We demonstrate that DG and its minimax part are able to detect various GAN failure modes (stable and unstable mode collapse, divergence, sample quality etc.) and rank different models successfully. Thus these metrics address two problems commonly faced by practitioners: 1) when should one stop training? and 2) if the training procedure has converged, have we reached the optimum? Finally, a significant advantage of the metrics is that, unlike many existing approaches, they require *no labelled data* and *no domain specific classifier*. Therefore, they are well-suited for applications of GANs other than the traditional generation task for images.

**Acknowledgements.**    The authors would like to thank August DuMont Schütte for helping with the ProgGAN experiment and Gokula Krishnan Santhanam and Gary Bécigneul for useful discussions.

## Footnotes

[2]The curves are obtained for a progressive GAN trained on CelebA

[3]For classification tasks using the zero-one test error is also very natural. Nevertheless, in regression tasks the test loss is often the only reasonable performance measure.

[4]Of course, one should also use a validation set, but this is less important for our discussion here.

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
