[Supplementary Material]

# A  Proof of Theorem 1

411 **Theorem 1** (DG and JSD). *Consider the GAN objective in Eq. [1], and assume that the generator*
412 *and discriminator networks have unbounded capacity. Then the duality gap of a given fixed solution*
413 $(G_{\mathbf{u}}, D_{\mathbf{v}})$ *is lower bounded by the Jensen-Shannon divergence between the true distribution $p_{\text{data}}$*
414 *and the fake distribution $q_{\mathbf{u}}$ generated by $G_{\mathbf{u}}$, i.e. $\mathrm{DG}(\mathbf{u}, \mathbf{v}) \geq \mathrm{JSD}(p_{\text{data}} \,||\, q_{\mathbf{u}})$. Moreover, if $G_{\mathbf{u}}$*
415 *outputs the true distribution, then there exists a discriminator $D_{\mathbf{v}}$ such that $\mathrm{DG}(G_{\mathbf{u}}, D_{\mathbf{v}}) = 0$.*

416 *Proof.* Let us denote by $p(x)$ the distribution over true samples and by $q_{\mathbf{u}}(x)$ the distribution over
417 fake samples generated by the generator $G_{\mathbf{u}}$. Let us also denote the output of the discriminator
418 by $D_{\mathbf{v}}(x)$. For simplicity, we will also slightly abuse notation and denote the GAN objective by
419 $M(q_{\mathbf{u}}, D_{\mathbf{v}})$. Thus, the GAN objective reads as follows,

$$M(q_{\mathbf{u}}, D_{\mathbf{v}}) := \frac{1}{2} \int p(x) \log D_{\mathbf{v}}(x) dx + \frac{1}{2} \int q_{\mathbf{u}}(x) \log(1 - D_{\mathbf{v}}(x)) dx . \tag{5}$$

420 **First we prove the first part of the proposition:**   Let us first recall the definition of the Jensen-
421 Shannon divergence of two distributions $p(\cdot), q_{\mathbf{u}}(\cdot)$,

$$\mathrm{JSD}(p \,||\, q_{\mathbf{u}}) := \frac{1}{2}\mathrm{KL}\left(p \,||\, \frac{p + q_{\mathbf{u}}}{2}\right) + \frac{1}{2}\mathrm{KL}\left(q_{\mathbf{u}} \,||\, \frac{p + q_{\mathbf{u}}}{2}\right) . \tag{6}$$

where the KL divergence is defined as,

$$\mathrm{KL}(p \,||\, q_{\mathbf{u}}) := \int p(x) \log\left(\frac{p(x)}{q_{\mathbf{u}}(x)}\right) dx .$$

422 Now given a fixed solution $(q_{\mathbf{u}}, D_{\mathbf{v}})$ we will show that the duality gap of this pair is bounded by the
423 Jensen-Shannon divergence. It is well known that this divergence equals zero if both distributions are
424 equal[5], and is otherwise strictly positive. To do so, we will first bound the minimax/maximin values
425 for $q_{\mathbf{u}}/D_{\mathbf{v}}$.

**(a)** Upper Bounding Minimax Value: Given $q_{\mathbf{u}}(x)$, the worst case discriminator is obtained by taking
the derivative of the objective in Equation (5) with respect to $D_{\mathbf{v}}(x)$ separately for every $x$ (this can
be done since we assume the capacity of $D_{\mathbf{v}}$ to be unbounded). This gives the following worst case
discriminator (see similar derivation in [9]),

$$D_{\mathbf{v}}^{\max}(x) := \frac{p(x)}{p(x) + q_{\mathbf{u}}(x)} .$$

426 Plugging the above value into Equation (5) gives the following minimax value,

$$\max_{D_{\mathbf{v}}} M(q_{\mathbf{u}}, D_{\mathbf{v}}) = M(q_{\mathbf{u}}, D_{\mathbf{v}}^{\max})$$
$$= \frac{1}{2} \int p(x) \log\left(\frac{p(x)}{q_{\mathbf{u}}(x) + p(x)}\right) dx + \frac{1}{2} \int q_{\mathbf{u}}(x) \log\left(\frac{q_{\mathbf{u}}(x)}{q_{\mathbf{u}}(x) + p(x)}\right) dx$$
$$= -\log 2 + \mathrm{JSD}(p \,||\, q_{\mathbf{u}}) \tag{7}$$

427 **(b)** Lower Bounding Maximin Value: Here we lower bound the maximin value for a given $q_{\mathbf{u}}(x)$,

$$\min_{q_{\mathbf{u}}} M(q_{\mathbf{u}}, D_{\mathbf{v}}) \leq M(p, D_{\mathbf{v}}) = \frac{1}{2} \int p(x) \log D_{\mathbf{v}}(x) dx + \frac{1}{2} \int p(x) \log(1 - D_{\mathbf{v}}(x)) dx . \tag{8}$$

Maximizing the last expression separately for every $x$ gives

$$\max_{D_{\mathbf{v}}(x) \in [0,1]} \frac{1}{2} p(x) \log D_{\mathbf{v}}(x) dx + \frac{1}{2} p(x) \log(1 - D_{\mathbf{v}}(x)) dx = -\log 2$$

428 Plugging the above into Equation (8) gives,

$$\min_{q_{\mathbf{u}}} M(q_{\mathbf{u}}, D_{\mathbf{v}}) \leq -\log 2 . \tag{9}$$

**(c)** Upper bound on Duality Gap: Recall the definition of Duality gap,

$$\mathrm{DG}(q_{\mathbf{u}}, D_{\mathbf{v}}) := \max_{D_{\mathbf{v}}} \mathrm{M}(q_{\mathbf{u}}, D_{\mathbf{v}}) - \min_{q_{\mathbf{u}}} \mathrm{M}(q_{\mathbf{u}}, D_{\mathbf{v}}) \ .$$

Using Equation (7) together with Equation (9) immediately shows that

$$\mathrm{DG}(q_{\mathbf{u}}, D_{\mathbf{v}}) \geq \mathrm{JSD}(p \,\|\, q_{\mathbf{u}}) \tag{10}$$

Therefore the duality gap is lower bounded by the Jensen-Shannon divergence between true and fake distributions, which concludes the first part of the proof.

**Next we prove the second part of the proposition:** Recall that we assume $q_{\mathbf{u}}(x) = p(x)$. And let us take,

$$D_{\mathbf{v}}(x) = \frac{1}{2}, \qquad \forall x$$

Next we show that the Duality gap of $(G_{\mathbf{u}}, D_{\mathbf{v}})$ is zero.

**(a)** Let us first compute the minimax value: Similarly to Equation (7) the following can be shown,

$$
\begin{aligned}
M(q_{\mathbf{u}}, D_{\mathbf{v}}^{\max}) &= \frac{1}{2} \int p(x) \log \left( \frac{p(x)}{q_{\mathbf{u}}(x) + p(x)} \right) dx + \frac{1}{2} \int q_{\mathbf{u}}(x) \log \left( \frac{q_{\mathbf{u}}(x)}{q_{\mathbf{u}}(x) + p(x)} \right) dx \\
&= -\log 2 \cdot \frac{1}{2} \int (q_{\mathbf{u}}(x) + p(x)) dx \\
&= -\log 2 \ .
\end{aligned}
\tag{11}
$$

where we used $p(x)/(p(x) + q_{\mathbf{u}}(x)) = \frac{1}{2}$.

**(b)** Let us now compute the maximin value. Since $D_{\mathbf{v}}(x) = \frac{1}{2}$ the following holds for any $q_{\mathbf{u}}^0(x)$,

$$M(q_{\mathbf{u}}^0, D_{\mathbf{v}}) = \frac{1}{2} \int p(x) \log D_{\mathbf{v}}(x) dx + \frac{1}{2} \int q_{\mathbf{u}}^0(x) \log(1 - D_{\mathbf{v}}(x)) dx = -\log 2 \ ,$$

which immediately implies,

$$\min_{q_{\mathbf{u}}} M(q_{\mathbf{u}}, D_{\mathbf{v}}) = -\log 2 \ . \tag{12}$$

Combining Equation (11) with Equation (12), with the definition of the Duality gap implies,

$$\mathrm{DG}(q_{\mathbf{u}}, D_{\mathbf{v}}) = 0 \ .$$

which concludes the second part of the proof.

$\square$

**Approximate computation of the duality gap** Note that the computation of the duality gap requires finding the exact solution for $\min_{q_{\mathbf{u}}} M(q_{\mathbf{u}}, D_{\mathbf{v}})$, and $\max_{D_{\mathbf{v}}} M(q_{\mathbf{u}}, D_{\mathbf{v}})$. In practice it is reasonable to assume that we may solve these two optimization problems only up to some approximation $\varepsilon$, i.e., that we may compute $q_{\mathbf{u}}^{*,\varepsilon}$ and $D_{\mathbf{v}}^{*,\varepsilon}$ such that,

$$M(q_{\mathbf{u}}^{*,\varepsilon}, D_{\mathbf{v}}) \leq \min_{q_{\mathbf{u}}} M(q_{\mathbf{u}}, D_{\mathbf{v}}) + \varepsilon, \qquad \& \qquad M(q_{\mathbf{u}}, D_{\mathbf{v}}^{*,\varepsilon}) \geq \max_{D_{\mathbf{v}}} M(q_{\mathbf{u}}, D_{\mathbf{v}}) - \varepsilon$$

In this case, a simple adaptation to the derivation above shows that the approximate duality gap computed using $q_{\mathbf{u}}^{*,\varepsilon}$ and $D_{\mathbf{v}}^{*,\varepsilon}$ gives us an upper bound on the approximate JS divergence as follows,

$$\mathrm{DG}_{\varepsilon}(q_{\mathbf{u}}, D_{\mathbf{v}}) := \mathrm{M}(q_{\mathbf{u}}, D_{\mathbf{v}}^{*,\varepsilon}) - \mathrm{M}(q_{\mathbf{u}}^{*,\varepsilon}, D_{\mathbf{v}}) \geq \mathrm{JSD}(p \,\|\, q_{\mathbf{u}}) - 2\varepsilon \ .$$

# B  Non-negativity of the practical duality gap.

While the DG is guaranteed to be non-negative in theory, this might not hold in practice since we do not have access to the exact $\arg\min$ and $\arg\max$ but instead optimize for a fixed number of steps. Therefore, a question that arises is whether the non-negativity property is affected by practical approximations of DG. First, note that negative DG values occur if at some step $t$, $M(\mathbf{u}_t, \mathbf{v}_{\mathrm{worst}}) < M(\mathbf{u}_{\mathrm{worst}}, \mathbf{v}_t)$. However, the optimization algorithm yields $M(\mathbf{u}_t, \mathbf{v}_{\mathrm{worst}}) > M(\mathbf{u}_t, \mathbf{v}_t)$ since $\mathbf{v}_{\mathrm{worst}}$ is initialized using $\mathbf{v}_t$ and optimized to maximize the objective. Similarly, we expect $M(\mathbf{u}_t, \mathbf{v}_t) > M(\mathbf{u}_{\mathrm{worst}}, \mathbf{v}_t)$, and DG is therefore non-negative. This of course assumes that the optimizer uses an appropriate set of parameters that guarantee successful decrease/increase of the objective. In Appendix D, we investigate the impact of the practical approximation of DG in lieu of the exact computation. In particular, we find that - both in theory and in practice - DG is not affected by the presence of mode collapse in $\mathbf{u}_{\mathrm{worst}}$.

|  | stable | | unstable | |
|---|---|---|---|---|
|  | lr G | lr D | lr G | lr D |
| RING | 1e-3 | 1e-4 | 1e-4 | 2e-4 |
| SPIRAL | 1e-3 | 2e-3 | 1e-4 | 2e-3 |
| GRID | 1e-3 | 2e-3 | 1e-4 | 2e-3 |

Table 5: Learning rates used for the toy experiments.

|  |  | Duality Gap | Modes | Std |
|---|---|---|---|---|
| stable | RING | 0.04 | 8 | 2375 |
|  | SPIRAL | 0.14 | 20 | 1999 |
|  | GRID | 0.03 | 25 | 2370 |
| unstable | RING | 13 | 2 | 152 |
|  | SPIRAL | 1.22 | 1 | 1724 |
|  | GRID | 12.09 | 3 | 37 |

Table 6: Final results for DG, number of covered modes and number of generated samples (out of 2400) that fall within 3 standard deviations of the means.

# C   Experiments

## C.1   Toy Dataset: Mixture of Gaussians

The toy datasets consist of a mixture of 8, 20 and 25 Gaussians for each of the models (RING, SPIRAL, GRID), respectfully. The standard deviation is set to 0.05 for all models except for the RING where the std is 0.01. Depending on the dataset, the means are spaced equally around a unit circle, a spiral or a grid.

The architecture of the generator consists of two fully connected layers (of size 128) and a linear projection to the dimensionality of the data (i.e. 2). The activation functions for the fully connected layers are relu. The discriminator is symmetric and hence, composed of two fully connected layers (of size 128) followed by a linear layer of size 1. The activation functions for the fully connected layers are relu, whereas the final layer uses sigmoid as an activation function.

Adam was used as an optimizer for both the discriminator and the generator with $beta_1 = 0.5$ and a batch size of 100. The latent dimensionality $z$ is 100. The learning rates for the reported models are given as follows in Table 5. The optimizer used for training the worst D/G is Adam and is set to the default parameters.

Plots of DG during training are given in Table 2. Table 6 lists the obtained results for the methods in terms of their final duality gap, number of modes they have covered and the number of generated points that fall within three standard deviations of one of the means. The heatmaps of the final generated distributions are given in Figure 9.

We also plot generated samples from the worst case generator in Figure 10.

Progress during training for Figure 12 is given in Figure 11.

Finally, the progress of DG, minimax, number of modes, and generated samples close to modes across epochs is given in Figure 12. High correlation can be observed, which matches the quantitive results in Table 7.

The correlation values are reported in Table 7. We observe significant anti-correlation (especially for minimax loss), which indicates that both metrics capture changes in the number of modes and hence *the minimax loss can be used as a proxy to determining the overall sample quality*.

## C.1.1   Loss Curves

A common problem practitioners face is when to stop training, i.e. understanding whether the model is still improving or not. See for example Figure 13, which shows the discriminator and generator losses during training of a DCGAN model on CIFAR10 [1]. The training curves are oscillating and

(a) Unstable ring    (b) Stable ring    (c) Unstable spiral    (d) Stable spiral    (e) Unstable grid    (f) Stable grid

Figure 9: Heatmaps of the generated distributions at the final steps. On top: trained model (stable or unstable), on bottom: $p_{\mathrm{data}}$

(a) Unstable ring

(b) Stable ring

(c) Unstable spiral

(d) Stable spiral

(e) Unstable grid

(f) Stable grid

Figure 10: Generated samples (in blue) from the worst generator for the discriminator for both the stable and unstable models. (ground truth in green color).

(a) Step0　　(b) Step5　　(c) Step10　　(d) Step15　　(e) Step20　　(f) Step25

Figure 11: Generated samples (blue) and real samples (green) throughout training steps

(a) DG　　　　(b) Minimax　　　　(c) Modes　　　　(d) Std

Figure 12: DG, minimax, number of modes, and generated samples close to modes across epochs

|  | DG | Minimax | |
|---|---|---|---|
| Modes | -0.63 | -0.97 | ring |
|  | -0.59 | -0.93 | spiral |
|  | -0.71 | -0.95 | grid |
| Std | -0.64 | -0.94 | ring |
|  | -0.64 | -0.58 | spiral |
|  | -0.7 | -0.93 | grid |

Table 7: Pearson product-moment correlation coefficients for an average of 10 stable rounds. Progress throughout the training of the individual metrics can be seen in Figure 12

hence are very non-intuitive. A practitioner needs to most often rely on visual inspection or some performance metric as a proxy as a stopping criteria.

The generator and discriminator losses for our 2D ring problem are shown in Figure 14. Based on the curves it is hard to determine when the model stops improving. As this is a 2D problem one can visually observe when the model has converged through the heatmaps of the generated samples (see Table 2). However in higher-dimensional problems (like the one discussed above on CIFAR10) one cannot do the same. Figure 15 showcases the progression of the duality gap throughout the training. Contrary to the discriminator/generator losses, this curve is meaningful and clearly shows the model has converged and when one can stop training, which coincides with what is shown on the heatmaps.

## C.2　Other hyperparameters

### C.2.1　Stable Mode Collapse

The architecture of the generator consists of 5 fully connected layers of size 128 with leaky relu as an activation unit, followed by a projection layer with tanh activation. The discriminator consists of 2 dense layers of size 128 and a projection layer. The activation function used for the dense layers of the discriminator is leaky relu as well, while the final layer uses a sigmoid. The value $\alpha$ for leaky relu is set to 0.3.

The optimizer we use is Adam with default parameters for both the generator, discriminator, as well as the optimizers for training the worst generator and discriminator. The dimensionality of the latent space $z$ is set to 100 and we use a batch size of 100 as well. We train for 10K steps. The number of steps for training the worst case generator/discriminator is 400.

We use the training, validation and test split of MNIST [6] for training the GAN, training the worst case generator/discriminator, and estimating the duality gap (as discussed in Section 3).

Figure 13: Discriminator and generator loss curves for a DCGAN model trained on CIFAR10. The curves are oscillating and it is hard to determine when to stop the training.

(a) Generator loss

(b) Discriminator loss

Figure 14: Discriminator and generator loss curves for the 2D ring problems. The curves are oscillating and it is hard to determine when to stop the training and when the model stops improving.

Figure 15: Curve of the progression of the duality gap during training.

Figure 16: Spikes in the DG progression curve reflect the quality of the generated samples

## C.3  3D GAN experiment

We train a simple GAN on a 2D submanifold mixture of seven Gaussians arranged in a circle and embedded in 3D space following the setup by [29]. The strength of the regularizer $\gamma$ is set to 0.1, and respectively, to 0 for the unregularized version.

Figure 16 shows a duality gap progression curve for a regularized GAN. It can be observed that the the spikes that occur are due to the training properties, i.e. at the particular step when there is an apparent spike, the quality of the generated samples worsens.

### C.3.1  Minimax experiment on Cifar10

Here we describe hyperaparameters for the experiment on Cifar10 [15]. The worst case discriminator we train is using the commonly used DCGAN architecture [27]. We again use Adam with the default parameters as the optimizer and the batch size is 100. We update the worst case classifier for 1K steps.

The hyperparameters used for the distortion in the experiment for visual sample quality are:

1. Gaussian noise
    (a) level 1: $sigma = 5$
    (b) level 2: $sigma = 10$
    (c) level 3: $sigma = 20$
2. Gaussian blur
    (a) level 1: $ksize = 2$
    (b) level 2: $ksize = 5$
    (c) level 3: $ksize = 7$
3. Gaussian swirl with strength 5
    (a) level 1: $radius = 1$
    (b) level 2: $radius = 2$
    (c) level 3: $radius = 20$

**Mode sensitivity and sample quality detection.**   Fig. 17 shows the ability of the three metrics (INC, FID and minimax) to detect mode dropping, mode invention and intra-mode collapse. We simulate mode dropping by using the class labels as modes. The input is a set of 5K images containing all 10 classes as 'real' images, and another set of 5K images composed of only subset of the modes (subset of 2, 4 and 8) as 'generated' images (Figure 17 a).

We then turn to *mode invention* where the generator creates non-existent modes. For this setting, the set of 'real' images contains only 5 classes, whereas the sets of 'generated' images are supersets of 5, 7 and 10 classes (Figure 17 b). *Intra-mode collapse* is another common issue that occurs when the generator is generating from all modes, but there is no variety within a mode. The 'generated' sets consist of images from all 10 classes, but contain only 1, 50 and 500 unique images within a class.

Figure 17 shows the trends for all metrics for the various degrees of mode dropping, invention and intra-class collapse. INC is unable to detect both intra-mode collapse and invented modes. On the other hand, *both FID and minimax loss exhibit desirable sensitivity to various mode changing.*

The ability of the metrics to react to distortion at an increasing intensity is in Fig. 18.

Figure 17: INC, FID and minimax loss for a) mode collapse (x-axis: how many modes out of 10 are generated); b) mode invention (x-axis: how many invented modes are generated) and c) intra-mode collapse (x-axis: number of unique images within a class). For INC higher is better; for FID and minimax lower is better.

Figure 18: INC, FID and minimax loss on samples at increasing intensity of disturbance

**Efficiency.** A metric needs to be computationally efficient in order to be used in practice to both track the progress during training and as a final metric to rank various models. Figure 19 shows the wall clock time in terms of seconds for all three metrics. We keep the number of update steps fixed to 1K which makes the computation of the minimax loss efficient, as are the other two metrics. The computation of DG takes twice the time of the computation of the minimax loss.

We also test the variance across rounds due to randomness in the seed and how this affects the final metric and overall ranking, on a simple mode collapse task. Table 8 summarizes the average of 5 rounds showing that the variance is negligible and does not affect the effectiveness of the metric.

### C.3.2 Experiment on cosmology dataset

Following the approach of [11], we used a Wasserstein loss with a gradient penalty of 10. Both the generator and the discriminator were optimized with an "RMSprop" optimizer and a learning rate of $3 \cdot 10^{-5}$. The discriminator was optimized 5 times more often than the generator.

Figure 19: Wall clock time (in log-scale) for the calculation of INC, FID and minimax loss for increasing number of samples to be processed

| classes | INC | FID | Minimax |
|---------|-----------|-------|-------------|
| 2 classes | $4.94\pm0.13$ | 63 | $-1.69\pm0.06$ |
| 4 classes | $7.84\pm0.3$ | 25.85 | $-2.53\pm0.04$ |
| 6 classes | $9.88\pm0.18$ | 18.39 | $-3.14\pm0.03$ |

Table 8:  Metrics on a simple mode dropping task

Real and generated samples are given side-by-side in Figure 20, showcasing the quality of the trained model.

### C.3.3   Experiment on text generation

For the implementation of SeqGAN, we used the TexyGen framework [35] and their preselected hyperparameters and metrics. Some generated samples are available in Tab9.

### C.4   Experiment on CIFAR10 for comparison of different GAN variants

We follow the setup and best hyperparameters reported in [16]. In particular, we train every model for 200K training steps and evaluate using 10K samples. For the computation of DG and minimax, we use 150 training steps and a train/test split of 9600/400 samples.

Figures 21, 22 and 23 show samples generated from the worst generator at various steps throughout training.

### C.5   Approximating the duality gap

We explored variants in which we are circumventing the optimization in order to find the worst case generator/discriminator by using a set of discriminators/generators out of which we choose the most adversarial one. The sets are created by saving snapshots of the parameters of the two networks during training. We explored variants where the snapshots come a) only from past models and b) a mix of previous and future models, and generally found b) to perform better. This setting is similar to the models proposed in [26], except they use skill rating systems to infer a latent variable for the

a man on his cell phone on ita cat is taking a picture of the door to the dark open sits .
a white bathroom with a black tub and a urinal and a toilet and a toilet .
a motorcycle parked on the sidewalk .
a motorcycle parked on a street looking up the street .
a living area is full wooden floor with plants growing fly to the toilet and a sinkairplane is , parked in a parking lot .

Table 9:  Generated samples from SeqGAN.

Figure 20: Real and generated cosmological images, representing slices of mass density of the universe. The yellow coefficient implies a high mass concentration

(a) Step 0    (b) Step 2K    (c) Step 4K    (d) Step 6K    (e) 8K

Figure 21: a-e: Generated samples at different training steps from the worst generator

successfulness of a generator. On the other hand, we compute the duality gap and the minimax loss to infer the successfulness of the entire GAN and of the generator, respectfully.

Table 10 gives the progression of the approximated duality gap for four various scenarios: stable model, unstable mode collapse and stable mode collapse. The duality gap was approximated using 10 models that spanned accross 2 epochs.

# D  Analysis of the quality of the empirical DG

The theoretical assumption appearing in the proof in Appendix A is that the discriminator and generator have unbounded capacity and we can obtain the true minimizer and maximizer when computing $\mathbf{u}_{\text{worst}}$ and $\mathbf{v}_{\text{worst}}$, respectively. This, however, is not tractable in practice. Furthermore, it is well known that one common problem in GANs is mode collapse. This raises the question of how

(a) Step 10K    (b) Step 12K    (c) Step 14K    (d) Step 16K    (e) 18K

Figure 22: a-e: Generated samples at different training steps from the worst generator

(a) Step 20K

Figure 23: a: Generated samples at the final step from the worst generator

Table 10: Progression of DG throughout training and heatmaps of the generator distribution. A1: stable ring, A2: unstable ring, B1: mode collapse, B2: stable mode collapse

the duality gap metric would be affected if the worst generator that we compute is collapsed itself. In the following we address this both empirically and from a theoretical perspective.

We use the same experimental setup as described in Appendix C.1. We focus on a GAN that has converged such that the generator covers all modes uniformly i.e. $p_g = p_{data}$ (Figure 24 a)). The discriminator outputs 0.5 for real and fake samples (Figure 24 b)). This means that the model has reached the equilibrium and the duality gap -in theory- is zero.

**Collapsed worst case generator.** Now we focus on the calculation of the duality gap. Let us consider the case of a mode collapsed worst case generator. In particular, when computing the maximin part of the duality gap i.e. $M(\mathbf{u}_{\mathrm{worst}}, \mathbf{v})$, let us assume the solution was such that $G_{\mathbf{u}_{\mathrm{worst}}}$ only covers one mode of the true distribution (Figure 24 d)). Then $M(\mathbf{u}_{\mathrm{worst}}, \mathbf{v}) = \log(0.5) + \log(1 - 0.5)$. The minmax calculation is: $M(\mathbf{u}, \mathbf{v}_{\mathrm{worst}}) = \log(0.5) + \log(1 - 0.5)$. Hence, the value of DG is zero, despite the collapse in the calculation for the $\mathbf{u}_{\mathrm{worst}}$. The generator has no incentive to spread its mass due to the objective. While this is a problem for the original GAN that is being trained, it is not an issue for the calculation of the duality gap metric.

Figure 25 b) shows samples generated from $G_{\mathbf{u}_{\mathrm{worst}}}$ when the experiment is performed in practice. We do observe that in this case, there is indeed a collapse that happened in the worst generator for

Figure 24: Analysis of a GAN that has reached the equilibrium for the mixture of 8 Gaussians problem. a) Samples generated from the GAN generator cover all 8 modes uniformly; b) Probabilities for a sample being real. The GAN discriminator assigns 0.5 probability to data points from the 8 modes and 0 everywhere else; c) For the computation of the duality gap, the theoretical $\mathbf{v}_{\text{worst}}$ assigns 0.5 to fake/real samples for the fixed GAN generator; d) We assume there was mode collapse when computing $\mathbf{u}_{\text{worst}}$ for the fixed GAN discriminator and samples from $G_{\mathbf{u}_{\text{worst}}}$ lie only on a single mode.

the fixed GAN discriminator. Yet, $D(G_{\mathbf{u}_{\text{worst}}}) = 0.489$ and $DG = 0.002$ confirming the previous thought experiment. A heatmap with generated samples from the GAN generator are given in Fig. 25.

Hence, mode collapse for the computation of DG is not an issue. Note though that when there is mode collapse in the GAN itself that is being evaluated, the DG detects this. In particular, this is supported by the high anti-correlation between DG and the number of covered modes and sample quality as shown in Table 7.

**Suboptimal solutions due to the optimization.** We now investigate the effect of the number of optimization steps used for the calculation of the duality gap on the quality of the solution. We run 5 different models with different hyperparameters with the goal to find the best setting. As suggested, we want to use the duality gap as the metric for this. Table 11 gives the results. The ranking of the models is the same for various numbers of optimization steps and corresponds to the ranking obtained by taking into consideration the number of covered modes and the number of generated samples that fall within 3 standard deviations of one of the modes.
This suggests that as long as one uses the same number of optimization steps when comparing different models, the suboptimality of the solution is empirically not an issue.

(a) GAN discriminator          (b) GAN generator

Figure 25: a) A heatmap of generated samples from the GAN generator (up) and the true data distribution (below). The generator is able to cover the true data distribution; b) Generated samples from $G_{\mathbf{u}_{\text{worst}}}$.

| hyperparameters | | quality of GAN | | DG for # optimization steps | | | |
|---|---|---|---|---|---|---|---|
| lr_D | lr_G | #modes (out of 8) | # quality samples (out of 2500) | 500 | 1000 | 1500 | 2000 |
| 2e-3 | 1e-4 | 8 | 2414 | 0.014 | 0.03 | 0.04 | 0.06 |
| 1e-4 | 1e-4 | 1 | 1119 | 10.02 | 11.3 | 12.23 | 12.3 |
| 1e-3 | 1e-4 | 8 | 2440 | 0.009 | 0.01 | 0.006 | 0.02 |
| 5e-3 | 1e-4 | 8 | 2478 | 0.008 | 0.002 | 0.002 | 0.001 |
| 1e-4 | 1e-5 | 1 | 501 | 10.84 | 12.37 | 13.25 | 13.7 |

Table 11: DG for various number of optimization steps and GAN hyperparameters. The set of the best hyperparameters is the same no matter the number of optimization steps are used for the calculation of the duality gap.

## Footnotes

[5]We mean equal up to sets of measure zero.