[Reviews · NeurIPS 2019]

Reviewer 1



- Clear and well presented paper - The experiment section is detailed and provides a lot of insights on the theoretical work of the duality gap. - The results are of significance and prove the value of the DP as a good measure

Reviewer 2



The idea of studying GANs from the game theory perspective is not new; however, using the duality gap as a performance metric (some sort of divergence between the generated data distribution and the real data distribution) is original to the best of my knowledge. The paper is written clearly. In terms of significance, while the idea of the duality gap is "natural" when considering the game theory perspective for GANs, it is not clear why this is a good metric for _any_ domain. The authors imply that it is a good idea to find a metric that does not depend on the domain of the data, but given all the parallels between GANs and the different divergences between probability distributions (JS, Wasserstein, etc.) I think the main problem is to find a metric that can be thought as correctly modeling the distance between high-dimensional datasets such as the ones given by images. In that case, modeling this aspect (which is highly domain-dependent) is crucial for understanding what a GAN is capturing about the data distribution. Of course the duality gap can be a good performance measure for certain domains, but I would argue that it depends heavily on the domain. Since game theory is well founded in clear assumptions, it is possible to find scenarios for which the duality gap is a good metric, scenarios for which one can test that the assumptions hold to a certain extent. However, this is not true in general. For images for instance, it is not clear why the duality gap is a good measure: What is the model that takes one to this conclusion? What are the assumptions? And, to which extent these assumptions are correct? I agree that finding methods that can be domain agnostic is one goal of ML research; however, for the particular problem of assessing the quality of a state-of-the-art generative model, I believe that understanding better how the networks involved actually encode the data in their weights is more important than one more performance metric besides FID and IS. Then again, the duality gap can be good for certain problems, but probably not for all, and at least for image generation it would only be one more metric, together with FID and IS. With all this said, I would argue that the significance of this work is good, but not great. Rebuttal update: The authors pointed out that the DG can be used for unlabeled datasets, which is an important remark that I took into consideration when reviewing the paper, and my score considers this property. The comparison with FID and IC was in the sense that there are no clear guarantees (or proper reasoning framework or model) for images (specifically) that this measure is somehow significant. The proposed guarantees come from game theory, but why is it a good framework for testing the quality of models of natural images? Still, I believe the paper is good when thinking about GANs in general (domain agnostic as the authors propose, for which the game theory framework makes sense). However, the fact is GANs are used in very specific settings (for modelling images mostly). Therefore, the significance of this paper is good, but not great. My score remains thus unchanged.

Reviewer 3



The proposed method is a nice contribution that provides a framework for evaluation of GANs using the duality gap theory. Specifically it considers the gap between various discriminator generator pairs (worst generator, discriminator). This can provide means for evaluation of gans in various settings. The method uses a good theoretical framework and well evaluated experimentally. It solves an important problem of evaluation of GANs

Reviewer 4



This submission aims to develop metrics for monitoring of GAN training and assessment of the quality for the resulting images. Such a metric is useful for inspecting the GAN convergence and to detect the mode collapse deficiency associated with GANs. This paper adapts the duality gap as the quality and provides a minmax approximation for the duality gap. Various experiments are performed that correlate the duality gap pattern with the convergence behavior. Strong points: -metrics for monitoring the GAN performance is novel and very practical -the experiments are extensive Weak points: -the approximation for the duality gap is rather ad hoc; it is not clear how close the approximation is to the real statistical gap; this needs further experimental exploration and justification -is the duality gap evaluated in parallel with the training? -the main result in this paper is Theorem 1. However, the usefulness of the lower bound for the duality gap is not immediately useful. It would be interesting if one can develop upperbounds.

[Author Response · NeurIPS 2019]



Figure 1: ProgGAN trained on CelebA: (left) losses vs. DG; (middle) gen. samples; (right) largest singular values of the conv layers

Figure 2: GAN trained on 1D Gaussian: (left) DG-approx vs. true DG; (middle) beginning of training; (right) end of training

We thank all the reviewers for their valuable input. We are pleased to see the positive feedback from Rev. 1, 2 and 3.
Rev. 5 highlighted some strengths in our submission while also having some comments which we address below.

## Reviewer 1:

**DG for face generation.** The DG curve in Fig. 1 from Sec. 1 is in fact created using a ProgGAN trained on CelebA
(see footnote on pg. 1). Due to space restrictions, we included this only in the introduction. We agree this can give a
more in-depth demonstration, and we plan to include it in the final version. Fig. 1 here shows the losses on CelebA,
generated samples, and the largest singular values of the conv. layers of G and D, which agree with the DG trend.

## Reviewer 2:

**"specify for which distributions DG is actually a good measure"** As you mentioned, each GAN objective (standard
GAN, WGAN) induces a different divergence (JS, Wasserstein), thus training a GAN optimizes the corresponding
divergence. It does not matter if the domain being considered is image, audio or text. As we show, in the case of
standard GANs, the DG measure is strongly related to the JS measure. We believe that a similar relation can also be
established for other GAN objectives (e.g. Wasserstein), which we will emphasize in the final version. Thus in terms of
theory, DG is indeed domain-agnostic. We also spent significant effort in the experimental section to validate that DG is
indeed a useful measure for toy problems, natural images, text, audio and cosmology data. For each domain, we report
solid levels of agreement with quality measures that are specific to each domain: FID and IS for natural images (where
they can be computed), time-frequency consistency for sound, nll for text, cosmo-score for cosmology...
**"For images DG would be yet another metric besides FID and IC"** Please note that unlike FID and IC, DG does
not require labeled data - hence DG can be easily computed for unlabeled datasets (e.g. CelebA) as well.
**"domain agnostic does not allow to better understand why and when (and how) GANS work"** The metric being
domain agnostic is a strength as it is very flexible and easily applicable. We show that DG can indeed help us understand
how and when GANs work (e.g. by analysing convergence - Fig. 2 or regularizers - Fig. 5). The need for such a metric
for further understanding GANs has been pointed out by many previous works e.g. see [Mescheder et al, ICML 2018].

## Reviewer 5:

**"is DG evaluated in parallel with the training?"** Yes. The computation is very fast (see Fig. 19).
**Guarantees.** We would like to highlight that we presented 2 performance measures. The first measure is DG which is
shown to be lower bounded by the JS divergence. While your question focuses on DG, the second measure provides a
different approximation which addresses this. This second measure is the Minimax loss which we define in Sec. 3. As
shown in Eq. 7 of the appendix: $minimax(u) = JS(q_u||p_{data}) - \log(2)$. Hence, minimax provides a direct handle to
the distance between true and fake distributions. This claim is also verified empirically (see Fig. 12, 17, 18).
**"the approximation for DG is rather ad hoc"** Please note that the approximation we use – which consists in taking
$n$ optimization steps instead of training till optimality – is very common in practice (eg. WGAN, WGAN-GP etc).
We verified the validity of our approximation in an extensive set of experimental results; in total we reported results
on 8 datasets belonging to 5 different domains (toy, natural images, text, cosmological data, audio). The outcome of
this study is that the approx. DG we calculate is sufficient to measure the performance of the training method, and
enables detection of convergence as well as different failure modes. We also provided experimental evidence that our
performance measure agrees with domain specific metrics (see Fig. 6, 7, 8).
**"this needs further experimental exploration"** Thank you for the suggestion. We did an additional experiment (Fig.
2 here) where we compare the approx. DG (i) *DG-approx* to (ii) *DG-true-grid* and (iii) *DG-true-conv*. The real data is a
1D Gaussian. Hence, for (ii) the true $G_{worst}$ can be computed using an extensive grid search within a wide interval,
whereas $D_{worst}$ is computed by optimizing till convergence. Similarly, for (iii) both $D_{worst}$ and $G_{worst}$ are optimized
till convergence, whereas (i) uses only a few steps. We see strong correlation- (i) and (ii):0.81, (i) and (iii):0.89 (ii) and
(iii):0.92. Finally, Section D "Analysis of the quality of the empirical DG" of the appendix further analyses exactly the
approximation quality. We will add a more prominent discussion and highlight this in the final version.

[Meta-Review · NeurIPS 2019]

The reviewers agree that this paper is novel, interesting and useful, and would make a useful addition to the literature. Please have a look at the updated reviews when preparing the camera-ready version, especially the comment on the usefulness of this metric for measuring the quality of generative models of images (the most popular domain for GANs).